# Infections of Tumor Prostheses: An Updated Review on Risk Factors, Microbiology, Diagnosis, and Treatment Strategies

**DOI:** 10.3390/biology12020314

**Published:** 2023-02-15

**Authors:** Andreas G. Tsantes, Pavlos Altsitzioglou, Dimitrios V. Papadopoulos, Drago Lorenzo, Carlo Luca Romanò, Thami Benzakour, Shinji Tsukamoto, Costantino Errani, Andrea Angelini, Andreas F. Mavrogenis

**Affiliations:** 1Microbiology Department, “Saint Savvas” Oncology Hospital, 11522 Athens, Greece; 2Laboratory of Haematology and Blood Bank Unit, “Attiko” Hospital, School of Medicine, National and Kapodistrian University of Athens, 12462 Athens, Greece; 3First Department of Orthopaedics, School of Medicine, National and Kapodistrian University of Athens, 12462 Athens, Greece; 42nd Academic Department of Orthopaedics, School of Medicine, National and Kapodistrian University of Athens, 14233 Athens, Greece; 5Clinical Microbiology, Department of Biomedical Sciences for Health, University of Milan, 20133 Milan, Italy; 6Studio Medico Associato Cecca-Romanò, 20121 Milano, Italy; 7Zerktouni Ortho Clinic, Casablanca 20250, Morocco; 8Department of Orthopaedic Surgery, Nara Medical University, Nara 634-8521, Japan; 9Department of Orthopaedic Oncology, IRCCS Istituto Ortopedico Rizzoli, 40136 Bologna, Italy; 10Department of Orthopaedics and Orthopaedic Oncology, University of Padova, 35122 Padova, Italy

**Keywords:** tumor prostheses, infections, diagnosis, management, prevention

## Abstract

**Simple Summary:**

Tumor prostheses are associated with a high infection risk. The main pathogens involved in these infections include *Staphylococcus* spp., followed by *Streptococcus* spp. These pathogens usually form a biofilm that covers the implants, which is a matrix of extracellular polymeric substances containing sessile microbial cells connected to one another. Prompt diagnosis is challenging due to the highly varying clinical symptoms and the lack of specific preoperative and intraoperative diagnostic tests. However, the development of molecular biology techniques such as polymerase chain reaction (PCR)-based methods and next-generation sequencing (NGS) over the past decade has improved the diagnostic accuracy for periprosthetic infections. Despite their high sensitivity, these methods have a high rate of false positive results. Surgical management with one- or two-stage revision surgery is mandatory for eradication of these infections. This study consolidates the current knowledge regarding the risk factors, microbiology, diagnosis, and treatment of infections of tumor prostheses.

**Abstract:**

Several causes contribute to the high infection rate in tumor prostheses, including extensive tissue dissection and patients’ immunosuppression due to the neoplastic disease. Most of these infections develop within the first 2 years following surgery with 70% of them occurring during the first year, while they are often associated with a low pathogen burden. The pathogenesis of infections in tumor prostheses is linked to bacteria developing in biofilms. Approximately half of them are caused by *Staphylococcus* spp., followed by *Streptococcus* spp., *Enterococcus* spp., and *Enterobacteriaceae* spp., while multiple pathogens may be isolated in up to 25% of the cases, with coagulase-negative *Staphylococci* (CoNS) and *Enterococccus* spp. being the most frequent pair. Although early detection and timely management are essential for complete resolution of these challenging infections, prompt diagnosis is problematic due to the highly varying clinical symptoms and the lack of specific preoperative and intraoperative diagnostic tests. Surgical management with one- or two-stage revision surgery is the mainstay for successful eradication of these infections. The recent advances in laboratory diagnostics and the development of biofilm-resistant prostheses over the past years have been areas of great interest, as research is now focused on prevention strategies. The aim of this study is to review and consolidate the current knowledge regarding the epidemiology, risk factors, microbiology, and diagnosis of infections of tumor prostheses, and to review the current concepts for their treatment and outcomes.

## 1. Introduction

Advances in imaging studies, a more thorough understanding of tumor biology, and improvements in adjuvant therapies have all contributed to the improved survival rates in patients with malignant bone tumors over the past decades [1]. Although limb amputation was once the norm for patients with bone tumors, currently, limb salvage surgery with megaprosthetic or allograft reconstruction is feasible in 80–85% of the cases [2]. Megaprosthetic reconstruction offers early skeletal stability, rapid rehabilitation, and improved functional outcomes, while patients’ acceptance and satisfaction rates are significantly higher compared to limb amputation [1,3]. However, the reported complication rate following implantation of tumor prostheses is 5–10 times higher than that of conventional total joint arthroplasties [4,5]. These complications include biological failures such as infection, aseptic loosening, and wound/soft tissue breakdowns, or mechanical failures such as implant breakage and instability. Unfortunately, most of these complications require additional procedures, while in many cases, revision of the implants is needed [6].

The infection rate after the index procedure ranges from 8% to 15%, while it can be up to 60% following revision surgeries [6]. The infection rate has been found to be comparable with respect to different limb-salvage reconstruction techniques, while the risk for amputation due to infection of tumor prostheses ranges from 23.5% to 87% [6,7,8]. These infections result in significant deterioration of patients’ health status, they increase the length of hospital stay and recovery period, while they are also associated with dismal functional outcomes and prognosis [7].

The vast majority of periprosthetic infections are due to direct inoculation intraoperatively or through hematogenous spread from a distant infected focus. All prosthetic implants remain susceptible to hematogenous seeding, especially during the first years following surgery due to the high vascularity of the periprosthetic tissues, which predisposes implants to a high infection risk.

The aim of this study is to review and consolidate the current knowledge regarding the epidemiology, risk factors, classification, and diagnosis of infections of tumor prostheses, and to review the current concepts on their treatment and prognosis.

## 2. Pathophysiology

Infections of tumor prostheses are often associated with a low pathogen burden and their pathogenesis involves bacteria developing in biofilms (Figure 1) [9]. Bacterial biofilm is defined as a sessile microbial community containing cells that are connected to a surface or to one another (embedded in a matrix of extracellular polymeric substances) and present with an exquisite phenotype in terms of growth rate and gene expression [7]. As opposed to bacteria developing in suspension cultures, biofilm bacteria have considerably higher antibiotic resistance.

The low organism burden and the presence of organisms in biofilms contribute to the poor Gram stain and culture sensitivity of synovial fluid specimens and periprosthetic tissue specimens. The most typical bacteria for these infections include *Staphylococcus* spp. (approximately 50%), followed by *Streptococcus* spp., *Enterobacteriaceae* spp., *Enterococccus* spp., *Pseudomonas aeruginosa,* and anaerobe species [7]. *S. aureus* is the most common pathogen that usually colonizes implants through the hematogenous spread from distant skin/soft tissues foci or through contiguous spread from local skin/soft tissue infections. Moreover, *S. pneumoniae* colonizes implants through hematogenous spread following respiratory tract infections, *Enterobacteriaceae* spp. (*Salmonella, Bacteroides, S. gallolyticus*) following gastrointestinal infections, while *E. coli* or *Klebsiella* spp. usually cause urinary tract infections and travel through hematogenous spread to adhere to implants. *Viridans Streptococci* usually cause dental procedure infections and colonize implants through hematogenous spread. The most common coagulase-negative *Staphylococcus* (CoNS) that causes periprosthetic infections is *S. epidermidis,* while *Staphylococcus lugdunensis* is another common CoNS. These pathogens are part of the normal skin flora of the perineum and hip regions. *Staphylococcus lugdunensis* mutates under stressful conditions, resulting in the formation of genetically modified small-colony variants (SCVs). The pathogenicity of S. lugdunensis is similar to that of S. aureus.

Moreover, multi-pathogen infections occur in approximately 25% of the cases, with coagulase-negative *Staphylococci* (CoNS) and *Enterococccus* spp. tending to be the most common pair. However, favorable treatment outcomes have not been linked to any specific pathogens [7,10].

## 3. Epidemiology

Tumor prostheses are associated with a higher infection rate compared to conventional arthroplasties due to the increased complexity of these reconstructive procedures, with the more extensive tissue dissection leading to larger bone/soft tissue defects, larger implants’ sizes, longer durations of these surgeries, higher numbers of previous surgeries, and the presence of malignancy leading to poor nutritional status and immunosuppression [7,11,12]. The infection rate following megaprosthetic reconstruction ranges from 2.2% to 34% [13,14,15,16], while the mean rate of periprosthetic infection is approximately 10% after the index surgery and 43% after revision surgeries [17,18]. Infection was identified as the most common reason for failure of endoprostheses by Henderson et al., with a reported rate of 7.8% [19]. In another study, infection-related revision surgery was required in 19.7% of proximal tibia replacements, and in 17.5% of total femur replacements [13].

## 4. Classification

Most infections occur within the first 2 years following surgery, while 70% of them develop within the first year. Periprosthetic joint infections have been classified as Class I (early or acute) when they develop within 4 weeks following surgery, Class II when they develop between 4 weeks and 2 years following surgery, and Class III (late or chronic) when they develop beyond that timeframe (Table 1). The majority of Class II infections are hematogenous.

Early postoperative and hematogenous infections are generally secondary to microorganisms of moderate virulence such as *Staphylococcus aureus*, and often develop with an abrupt onset of clinical symptoms such as wound leakage, acute fever, discomfort, swelling, and effusion along with erythema around the incision site [11,14]. Untreated early infections lead to the development of chronic infections with sinuses, bacteremia, and sepsis. Late postoperative periprosthetic joint infections are usually low-grade infections and more frequently linked to pathogens of low virulence, such as Coagulase-Negative *Staphylococccus* [13]. Late infections present with more subtle signs such as persistent chronic inflammation and postoperative discomfort. Although rapid loosening of the prosthetic implants is usually indicative of infection, radiographic signs of early loosening are common; thus, differential between septic and aseptic loosening can be challenging, unless a discharging sinus is present [14].

## 5. Diagnosis

Early diagnosis is paramount for the successful treatment of periprosthetic infections following implantation of tumor prostheses (Table 2). However, due to the highly varying symptoms and the large number of nonspecific laboratory tests, diagnosis is often challenging.

A combination of clinical, histological, and microbiological parameters is recommended in order to increase the diagnostic capacity for these infections, and to rule out other causes of periprosthetic pain. Diagnosis is confirmed in cases of pus development, sinus discharge, and/or a positive cultures [13]. Antibiotics should be halted until all microbiological testing is completed unless prompt antibiotic therapy is mandatory due to developing signs of sepsis. Infection may be suspected in cases of elevated C-reactive protein (CRP) levels or elevated white blood cell count (WBC) [14]. However, elevated CRP levels and WBC count are common during the early postoperative period; therefore, these studies are not helpful during the first 2 weeks after surgery; following that period, consistently high levels are more indicative of an infection [14].

Imaging evaluation of periprosthetic infections include simple radiographs, bone scintigraphy, ultrasonography, and magnetic resonance imaging (MRI). Although radiographs are helpful in ruling out other causes of periprosthetic pain, they are neither sensitive nor specific for the early diagnosis of a periprosthetic infection. Radiographs may reveal bone loss and implant loosening; however, these findings are not specific for infection, and they may be due to aseptic implant loosening. Ultrasonography may reveal joint effusion or synovial hypertrophy, while MRI is usually not helpful, due to artifacts from the presence of implants. Bone scintigraphy with the combined use of labeled leukocytes and Technetium 99m-sulfur colloid, Gallium citrate 67, or Indium 111 can aid in the diagnosis of periprosthetic tumor infections, with a reported accuracy of 88–98% [20]. In the case of bone infections, leukocytes accumulate in the bone marrow due to phagocytosis by the reticuloendothelial cells. As opposed to labeled leukocytes, sulfur colloid does not accumulate in bone marrow in cases of infection; therefore, an increased activity on the labeled leukocyte imaging without equivalent activity with Technetium 99m-sulfur colloid increases the detection capacity for infection [21]. Moreover, the combined use of labeled leukocytes with Technetium 99m-sulfur colloid is helpful in distinguishing an infection from aseptic loosening [22,23,24]. Aseptic loosening is typically followed by a vigorous immune response, which induces transformation of fatty marrow into hematopoietically active marrow, resulting in an increased activity on the bone marrow imaging with Technetium 99m-sulfur colloid [25].

Joint aspiration is recommended, and the synovial fluid should undergo extensive histological and microbiological testing. However, joint aspiration is not always feasible, while a negative result does not rule out periprosthetic infection; cultures may be negative due to prior use of antibiotics, the lack of planktonic pathogens due to their adherence to the implant surface, and inappropriate culture media such as in cases of anaerobic bacteria or fastidious/atypical organisms (mycobacteria, etc.). Sinus tract cultures should be avoided due to the high contamination rate by the normal skin flora. Traditionally, tissue cultures have been considered the most reliable method for identifying the causative pathogens, confirming periprosthetic infections. However, the sensitivity of tissue cultures ranges from 70% to 90%, while the specificity ranges from 67% to 91% [26,27,28]. Biopsy from multiple sites is recommended in order to increase the detective accuracy of the obtained tissue cultures [11].

Over the past decade, molecular biology techniques have gained ground for diagnosis of periprosthetic infections. These techniques mainly include polymerase chain reaction (PCR)-based methods and next-generation sequencing (NGS). PCR is a robust diagnostic modality that detects DNA traces of pathogens and can be applied to various specimens such as synovial fluid, tissue samples, or fluid from sonicated implants [29,30,31]. There are two different methods of PCR technology: a commercial kit or a house-made methodology. The development of multiplex commercial PCR kits has limited the use of house-made methods. The diagnostic accuracy of PCR in terms of sensitivity and specificity is higher than that of tissue cultures, while another advantage of PCR is that it can detect all-existing bacteria. Broad-range PCR that targets the 16S gene is an alternative to the traditional quantitative PCR. While traditional quantitative PCR detects specific pathogens based on the used primers, broad-range 16S PCR is a non-specific technique that detects the presence of any bacterial DNA in the specimen. The sensitivity of broad-range 16S PCR varies from 50% to 92%, while the specificity ranges from 65% to 94% [32]. An important limitation of all PCR techniques is contamination, which leads to a high rate of false positive results, as PCR cannot distinguish between dead or living bacteria DNA. Therefore, proper quality measures with great caution regarding the sterile collection of biological samples for analysis and isolation of DNA are of great significance [33,34]. The second molecular biology technique that has been increasingly used over recent years is NGS. DNA sequencing allows for the detection of a large range of pathogens, including not only bacteria, but fungi as well [35]. This method completely disconnects laboratory evaluation from a culture-based approach, while its sensitivity is significantly higher than that of any other method [36,37,38]. Moreover, in theory, contamination is not an issue, as PCR is not required. However, the improved sensitivity is counterbalanced by the lower specificity value of NGS, resulting in false positive results [36].

Sonication of the implants and culture of the sonication fluid is another technique for the diagnosis of periprosthetic infections. Most studies regarding sonication include patients following conventional total joint arthroplasties, and evidence regarding this diagnostic modality in tumor protheses is scarce. Sonication fluid cultures have been shown to have higher sensitivity compared to tissue cultures, especially in patients who received antibiotics preoperatively [39,40,41,42]. Sambri et al. evaluated the detective accuracy of tissue cultures vs. sonication fluid cultures for the diagnosis of infections of tumor protheses and found that the sensitivity and the negative predictive value were significantly improved for sonication fluid cultures, whereas specificity and positive predictive value were comparable between tissue cultures and sonication fluid cultures [43]. The authors of this study concluded that a sonication fluid culture should be used in addition to the Musculoskeletal Infection Society’s diagnostic criteria.

## 6. Risk Factors

The location of the tumor has been shown to be related to the infection rate in tumor protheses; studies have reported that lesions in the proximal tibia, proximal femur, and pelvis are associated with an increased risk for periprosthetic infections, while lesions in the distal femur and proximal humerus are related to a lower infection rate [44]. Moreover, knee periprosthetic joint infections have worse prognosis [45,46,47].

Procedure-related factors associated with a higher infection risk include >37% resection of the proximal tibia, resection of >2 heads of the quadriceps muscle, longer duration of surgery (>2.5 h), and transfusion with >1 unit of concentrated red blood cell [48]. Moreover, sufficient soft tissue coverage is essential for prevention of infections, and a lower incidence of infections has been reported with the routine use of a gastrocnemius flap following proximal tibia resections [46]. In one study evaluating patients who underwent limb salvage surgery and prosthetic reconstruction, the infection rate in patients who did not receive radiation therapy was 9.8%, compared to 20.7% in those who received preoperative radiation therapy and 35.3% in those with postoperative radiation therapy [49]. Preoperative hospitalization > 48 h, malnutrition, anemia, chemotherapy, male gender, older age, diabetes, previous native joint infection, obesity, skin diseases, and admission to the intensive care unit have also been identified as significant risk factors for infections in tumor prostheses [48,50]. According to Meijer et al., certain laboratory parameters such as lower preoperative hemoglobin and albumin levels are associated with a higher infection rate following reconstructive surgery for proximal humerus tumors, and the authors recommended that these parameters should be optimized prior to surgery [51].

Metastatic spread of the tumor and its histological characteristics were not found to be related to the incidence of infections in most studies, while there is an ongoing debate about whether primary or metastatic lesions are more likely to develop infections [44,45,46,47,48,49,50,51,52]. Last, although most studies report similar infection rates among different types of tumor protheses, there is some evidence that cemented prostheses have a higher infection rate compared to uncemented prostheses [7,46,52,53,54,55].

## 7. Duration of Surgery

Numerous studies have shown that longer durations of surgery in total joint arthroplasties are associated with a higher infection risk [56,57,58,59]. In a comprehensive review by Cheng et al., the infection risk was found to be roughly doubled in surgeries lasting >1–4 h, and nearly tripled in procedures lasting >5 h [60]. Moreover, the risk of surgical site infections was found to be increased by 5% every 10 min, 13% every 15 min, 17% every 30 min, and 37% every 60 min. However, this study was limited by the high heterogeneity in the included patients, and the high heterogeneity regarding the criteria used for the definition of periprosthetic joint infections. In a single-center study, Peersman et al. reported that the risk of infection was considerably increased when the duration of total knee arthroplasty was more than 2.5 h [61].

As opposed to the findings of the previous studies, Pulido et al. evaluated 9,245 patients following total hip and knee arthroplasties and reported that the duration of surgery was not independently associated with the development of periprosthetic joint infections [62]. In line with the findings of Pulido et al., several other studies have also failed to demonstrate a positive correlation between duration of surgery and development of infections, while even an inverse relationship was shown in some of them [63,64,65,66,67].

## 8. Antibiotic Prophylaxis

While there are widely accepted protocols for antibiotic prophylaxis in conventional total joint arthroplasties, there are no respective protocols in orthopedic tumor surgeries. Parenteral preoperative antibiotics have been shown to lower the infection rate following conventional total joint arthroplasties [18]. In a meta-analysis including seven studies with 3065 participants who underwent total joint arthroplasties, the relative risk of infection was reduced by 81% in patients receiving antibiotic prophylaxis compared to placebo [68]. However, prophylaxis in tumor patients undergoing limb salvage surgery and prosthetic reconstruction was not evaluated in any of the studies included in this meta-analysis.

The optimal duration of perioperative antibiotics for the prevention of infections in tumor prostheses has not been elucidated yet. Although a long course of postoperative antibiotics has been the standard of care in high-risk patients, there is no sufficient evidence supporting the effectiveness of this approach compared to the lower duration of antibiotic prophylaxis. Although there is a general agreement that postoperative antibiotics should not be given for >24 h postoperatively after primary total joint arthroplasties, this is debatable in orthopedic tumor surgeries. Most clinical guidelines recommend the perioperative use of antibiotics in total joint arthroplasties and other orthopedic procedures involving implants [69,70,71,72]. The administration of one dose preoperatively and for the following 24 h postoperatively was by the International Consensus Meeting on Periprosthetic Infections [70]. However, the Centers for Disease Control and Prevention Guidelines recommend only one dose preoperatively and they do not recommend redosing in patients undergoing total joint arthroplasties [71]. On the other hand, the American Association of Hip and Knee Surgeons (AAHKS) advises that postoperative antibiotics should be continued for 24 h and recommends that further research should be performed to ascertain whether a shorter duration of antibiotic prophylaxis is safe and effective [69]. The American Academy of Orthopaedic Surgeons (AAOS) released its own recommendations, which are in line with those of AAHKS [72] (Table 3).

The Prophylactic Antibiotic Regimens in Tumor Surgery (PARITY) study is an ongoing worldwide randomized controlled trial and expects to enroll 600 patients by the end of the year (NCT01479283) [73]. In the subgroup population of orthopedic tumor patients, this study will evaluate whether 5 days of postoperative antibiotics is superior to one day of postoperative antibiotics in terms of decreasing the postoperative infection rate. The rate of postoperative infections during the first year will be the main result, while secondary outcomes will include antibiotic-related adverse events, patient functional results, quality-of-life ratings, reoperation rates, and mortality rates.

Intraoperative redosing is usually recommended to ensure adequate serum and tissue antibiotic concentration if the procedure lasts for more than two half-lives of the antibiotic, or if there is excessive blood loss (>1500 mL) [70]. The redosing interval should start being counted from the time of the preoperative dose, rather than the onset of the procedure. Although multiple-dose antibiotic prophylaxis extending over 24 h is a common practice in tumor surgeries, a recent study evaluating patients undergoing implant-based breast reconstruction showed that there were comparable rates of postoperative infections between single vs. multiple doses of prophylactic antibiotics [74]. However, based on the current guidelines, most scientific organizations recommend multiple-dose antibiotic prophylaxis for at least 24 h [69,70,72]. Notably, these guidelines refer to conventional joint arthroplasties and not to tumor reconstructions; therefore, the need for long-term vs. short-term antibiotic prophylaxis in orthopedic tumor surgeries should be investigated in future studies.

The appropriate antibiotic prophylaxis may differ in tumor patients compared to total joint replacements as these patients have markedly different characteristics [73]. In a systematic review by Racano et al. including patients with lower-extremity tumors who underwent surgery and prosthetic reconstruction, the authors found that long-term antibiotic prophylaxis was more effective than short-term prophylaxis (pooled weighted infection rate, 8% vs. 13%) in reducing the infection rate [18]. The results of this study highlight the different characteristics of tumor patients compared to patients undergoing conventional total joint arthroplasties, underlining the need for the development of specific infection control strategies in tumor surgeries.

High-quality studies are lacking regarding antibiotic prophylaxis in patients who had preoperative radiotherapy, soft tissue or bone resection, and prosthetic or allograft reconstruction. Most studies do not report sufficient data regarding the antibiotic regimens (dosage, duration, etc.), while the heterogeneity in the patient groups included makes even more problematic the comparison of the infection rates between different perioperative antibiotic regimes. Therefore, high-quality, randomized controlled studies are necessary to compare the effectiveness of various antibiotic regimens in these patients.

The National Comprehensive Cancer Network (NCCN) and the Infectious Diseases Society of America (IDSA) guidelines recommend prophylaxis with fluoroquinolones in high-risk patients with neutropenia. These recommendations have also been supported by several meta-analyses, which mostly focused on patients with hematological malignancies [75,76]. However, antibiotic prophylaxis in patients with primary bone tumors or patients with a tumor prosthesis was not evaluated in any of the studies included in these meta-analyses. Based on the IDSA guidelines, patients with neutropenia predicted to resolve within 7 days, no active medical comorbidity, and stable hepatic/renal function are classified as “low-risk patients”; most patients with solid tumors are considered “low-risk” based on the IDSA definition. The use of antibiotic prophylaxis in low-risk patients is debatable due to the emergence of bacterial resistance, and the questionable effectiveness of fluoroquinolones as a preventative measure for infections, compared to their use as a treatment for infections [77]. Although it has been shown that the use of fluoroquinolone prophylaxis does not alter the incidence of infections brought on by resistant bacteria, there is still substantial concern about the development of bacterial resistance due to the uncontrolled use of antibiotics. Based on these recommendations, individuals with a tumor prosthesis should not typically receive antibiotic prophylaxis during neutropenic episodes.

Prophylaxis should be administered in line with the current recommendations for conventional arthroplasties as there are no high-quality studies evaluating antibiotic prophylaxis in patients undergoing tumor surgery and prosthetic reconstruction pre- or post-radiation therapy or chemotherapy [69,70]. However, it is recommended to administer prophylactic antibiotics for longer than 24 h in tumor surgeries, while in conventional arthroplasties, antibiotics are recommended to be administered for no more than 24 h [18].

## 9. Chemotherapy and Radiation Therapy

Chemotherapy has been shown to increase the overall revision rate of prosthetic reconstruction by 30%, as a result of the decreased osseointegration of the implants [78]. However, it is yet unknown whether chemotherapy affects the infection rate following prosthetic reconstruction and whether immunodeficiency due to chemotherapy poses a risk factor for infection. Although some studies have shown a causal association between chemotherapy and infection in tumor patients [50,70,79,80,81], a large body of evidence supports that chemotherapy does not significantly alter the incidence of infections in these patients [82,83]. Miwa et al. compared the infection rates between patients with and without adjuvant treatment (such as chemotherapy), and found no difference between the two groups [84]. On the other hand, Kapoor et al. reported a 20% higher infection rate due to the reduced immune response after neoadjuvant chemotherapy [16].

The results of the literature regarding the association between adjuvant therapy (chemotherapy or radiation) and the risk of postoperative infection in patients with bone tumors and metastatic bone disease differ based on the type and location of the tumor. Chemotherapy or radiation therapy did not increase the risk of infection, according to a study evaluating patients who underwent various lower-extremity tumor surgical operations, while chemotherapy was also not a risk factor for infection in two other studies including patients who underwent surgery for primary bone tumors in the lower extremities, and prosthetic reconstructions for knee tumors [13,84,85]. However, preoperative radiation was found to be associated with a greater incidence of wound complications in some studies, while postoperative radiation therapy was also linked to an increased risk of infection in patients with spinal metastases [19,86,87,88,89]. Some infections following adjuvant chemotherapy could be treated with antibiotics and modification of the myelosuppression therapy, while, in some cases, a revision surgery was needed [90].

Chemotherapy alters the gut flora and results in chemotherapy-associated mucositis, while the skin’s microbiome is also altered by chemotherapy leading to frequent fungal infections in tumor patients [91,92,93,94]. The association between postoperative prosthetic infections and changes in the gut and skin microbiomes following chemotherapy has not been fully elucidated. It has been shown that the baseline diversity in stool bacteria of patients who developed infections following chemotherapy for acute myeloid leukemia was considerably lower [95]. Moreover, chemotherapy has been shown to induce changes in the gut microbiome of patients with lymphoma, such as an increase in *Proteobacteria* spp. (*Escherichia*, *Salmonella*, *Vibrio*, *Helicobacter*, *Yersinia*, *Legionellales*) [96]. The prevalence of *Proteobacteria* spp. in the gut microbiome prior to chemotherapy was found to be associated with the infection risk [96]. Moreover, reduced diversity in the taxa of the gut microbiome has also been linked to the risk of bloodstream infections following chemotherapy [97]. However, there are no recommendations regarding the evaluation of the gut’s microbiome following (neo-) adjuvant chemotherapy.

The association between development of infections and changes in the gut/skin microbiome due to chemotherapy is less likely in periprosthetic infections, as these infections are usually caused by different microorganisms (e.g., *Staphylococcus, Streptrococcus, Enterococcus,* and *Pseudomonas* species) than those that have been noted to be altered due to chemotherapy (e.g., *Proteobacteria* and *Fungi*) [14,92,98,99]. Moreover, any changes in the gut/skin microbiome due to chemotherapy have been typically restored by the time periprosthetic infections develop, as these infections typically develop far later than the end of chemotherapy [13].

The optimal duration of the time period between surgery following preoperative radiation therapy and/or chemotherapy is still debatable. To date, the decision for the time of surgery after neoadjuvant treatment is based on each hospital’s protocol and physicians’ preference. There is only scarce evidence regarding the association between the incidence of infection and the time interval between neoadjuvant therapy and prosthetic reconstructions in tumor patients. There are numerous mechanisms through which radiation hinders wound tissue healing; ionizing radiation can harm fibroblasts, resulting in sluggish development, dermal atrophy, necrosis, and ultimately decreased wound tissue strength [99,100]. Therefore, surgery is recommended to be avoided immediately after radiotherapy, and a waiting period of 4 weeks is usually needed for the restoration of the normal tissue’s properties [101]. Similarly, systemic side-effects of chemotherapy such as toxicity and immunosuppression are very well known, and the decision for surgical resection of a tumor following chemotherapy is still based on the treating physician’s preference.

The impact of preoperative radiation on wound complications was studied by Keam et al. who did not find any difference in the complication rate when surgery was performed before and after 30 days of radiation [102]. Griffin et al. evaluated the impact of different time intervals (3 weeks, 4 weeks, 5 weeks, 6 weeks, and ≥6 weeks) between preoperative radiotherapy and surgery on the wound complication rate [100]; although the rate of wound complications was higher at the time points of 3 weeks and ≥6 weeks (34–38%) compared to the time points of 4–6 weeks (28–31%), there was no statistical difference between these time points [100].

## 10. Neutrophil Count

WBC count and absolute neutrophil count (ANC) should be evaluated prior to limb salvage surgery and prosthetic reconstruction. Due to the high complexity of limb salvage surgery, all preventive measures should be considered in order to decrease the risk of infection. Neutropenia is defined as ANC ≤ 1500/mm^3^ and this cutoff value has been historically considered as a significant risk factor for the development of infections and of other detrimental complications. This association was initially described by Bodey et al. who reported that the infection rate ranged from 14% in patients with ANC < 1000/mm^3^ to 60% in those with ANC < 100/mm^3^, and the risk was increasing as ANC was declining [103]. This association was also seen in a more recent study by Lima et al. who evaluated patients with ANC < 500 cells/mm^3^ and found that the infection rate was also higher in patients with a lower ANC [104].

Neutropenic patients undergoing abdominal surgery were assessed by Natour et al. who evaluated patients with ANC < 500/mm^3^, between 500 and 1000/mm^3^, and between 1000 and 1500/mm^3^ [105]. The authors of this study reported that a higher incidence of surgical site infections, longer hospital stay, and higher mortality rate were seen in patients with lower ANC. In line with the results of the previous study, Gulack et al. also found that leukopenia prior to urgent abdominal surgery was a significant predictor of postoperative morbidity and mortality [106]. However, the authors of this study did not find any significant difference in the incidence of deep wound infection between leukopenic patients and those with a normal preoperative WBC count. In another recent study, the infection rate was higher in cancer patients with low ANC compared to those with normal ANC following implantation of a port device [107]. Although there are no studies evaluating the association between the infection risk and ANC following limb salvage surgery, it is recommended that patients with ANC < 1000/mm^3^ due to chemotherapy or to the neoplastic disease should not undergo surgery until ANC increases >1000–1500/mm^3^ [95,97].

## 11. Surgical Drains

Although evidence supporting the beneficial use of surgical drains in most surgeries is lacking, orthopedic surgeons commonly use drains, especially in tumor surgeries [108,109,110,111]. On the other hand, there are some indications that surgical drains are associated with an increased risk of infection following musculoskeletal tumor surgeries. The key drawback regarding the use of a surgical drain is the development of a communication pathway for bacteria between the deep tissue layers and the open environment; therefore, there is an increased likelihood for pathogens such as *Staphylococcus aureus* to contaminate the deep wound tissues [112]. Surgical drains are usually left in place for 2–3 days, while a shorter period of drain use has not been evaluated regarding its association with a higher infection risk in musculoskeletal tumor surgeries.

Parker et al. conducted a large meta-analysis of randomized controlled studies regarding the use of surgical drains in orthopedic surgeries [113]. The authors of this meta-analysis did not observe any difference in the infection rate, the incidence of wound hematomas, or the need for additional surgery, due to wound complications between patients with and without surgical drains. However, blood transfusions were more common in patients in whom drains were used. The same results have also been shown in numerous other studies including patients undergoing conventional total joint arthroplasties, general surgery procedures, and orthopedic trauma procedures [114,115,116]. A Cochrane Systematic Review of 36 trials including 5464 patients and 5697 surgical wounds was also conducted in order to evaluate whether surgical drains are associated with a higher infection rate [117]. Although tumor patients were not independently evaluated in the review, there was no difference between patients with and without drains in terms of wound infection, hematoma, dehiscence, or reoperation rates. Specifically, the incidence of surgical site infection was 1.9% in patients who had a closed suction drain compared to 2.4% in those without a drain. Blood transfusions were also more common in those who had drains [112].

Rossi et al. reported an overall infection rate of 8.7% in a study including 723 musculoskeletal tumor procedures when drains were used for 2–3 or 5 days (for non-pelvic tumors and pelvic tumors, respectively) [109]. Currently, there is no agreement on the optimal duration of surgical drains following surgery. According to the World Health Organization (WHO), low-quality evidence indicates that early removal of surgical drains does not affect the infection rate compared to late removal [118]. In a retrospective analysis of 165 patients who had musculoskeletal tumor surgery in whom two surgical drains were used for 24 h, Lerman et al. reported an infection rate of 10.3% [118]. Similarly, Shehadeh et al. reported a 11.3% infection rate with the use of surgical drains [6]. In another meta-review including 34 systematic reviews, the association between drains and infection risk following various surgical operations was evaluated [118]. Although there was no statistical significance, a trend toward a decreased incidence of wound infections was found when no drain was used. Patel et al. reported a 42% relative risk increase in surgical site infections, with every additional day of surgical drain use [112]. On the other hand, Sankar et al. in their prospective study including 214 orthopedic surgical operations did not observe any association between the length of drain use and development of wound infection [115]. In another study, drain sites were swabbed following drain removal, and the drain tips were sent for culture [119]. Although the authors of this study reported that the risk of bacterial colonization rose over time, this was not necessarily associated with the development of clinical infection. In line with the findings of the previous study, Willett et al. reported a close relation between the duration of surgical drain retention and the incidence of positive cultures; however, the clinical infection rate was not increased [120]. Additionally, there is no evidence supporting the use of antibiotics until surgical drain removal, although this has not been evaluated in tumor surgeries [121,122].

## 12. Cemented vs. Cementless Fixation

The association between the infection rate and the type of fixation (cemented vs. cementless) has been investigated in numerous studies [4,123,124,125,126,127,128,129]. The use of antibiotic-loaded bone cement has now become the standard in daily clinical practice. However, the results of the literature are highly varying regarding the association between the infection rates and the type of fixation in musculoskeletal tumor surgeries. Infection rates in cemented megaprostheses range from 5.2% to 21.9% [4,123,124,125,126], while the infection rates in cementless megaprostheses range from 9.7% to 12% [127,128,129].

The association between infection rate and type of fixation was evaluated in a systematic review of 40 studies including patients with proximal tibial and distal femoral replacements; in this review, the infection rate was higher in the cemented distal femoral replacements compared to the cementless (9.0% vs. 5.8%) [15]. According to linear regression analysis, the risk of infection significantly increased over time for cemented distal femoral replacements but did not increase for cementless distal femoral replacements. As opposed to distal femoral replacements, the infection rate was similar between cemented and cementless proximal tibial replacements (15.2% vs. 14.1%); the risk of infection in proximal tibial replacements did not increase over time [15]. Pala et al. evaluated the infection incidence in cemented and cementless megaprostheses in the lower extremities, [130] and reported that the survival rate at 60 months was 68% for cemented compared to 82% for cementless megaprostheses. The authors of this study concluded that cementless fixation was associated with increased survival rate.

## 13. Surgical Treatment

### 13.1. Irrigation and Debridement

There are no guidelines regarding the optimal surgical treatment of infections following implantation of tumor prostheses. The effectiveness of a protocol including debridement, antibiotics, and implant retention (DAIR) in infected megaprostheses has been evaluated in numerous studies and the success rate of this protocol ranges from 39% to 70% [16,47,54,131,132,133]. Dhanoa et al. in their study of 105 infected tumor prostheses supported that debridement could be performed if the following requirements are met: acute onset of infection postoperatively (14–28 days), confirmed diagnosis based on histopathology and microbiology, stable implants, and susceptibility of the isolated pathogen to an efficient and orally available antimicrobial agent [54]. The authors of this study reported a 42.8% infection eradication rate of a DAIR protocol when these criteria were met. The authors of this study also recommended repeat debridement in patients in whom the initial debridement failed; in these cases, a two-stage revision surgery was advised. However, Kapoor et al. recommended an additional debridement following a failed surgical debridement, before switching to a two-stage revision surgery [16]. Superficial infection, short duration of symptoms, well-fixed implants, and isolation of a highly susceptible pathogen have been documented as prognostic factors for improved outcomes after debridement in multiple studies [131,132,133,134].

Allison et al. [134] evaluated different treatment strategies for infections in tumor prostheses and reported a 42% eradication rate with irrigation and debridement without implant exchange, a 70% eradication rate after single-stage debridement and exchange of the modular components, and a 62% eradication rate after a 2-stage revision surgery. However, Jeys et al. reported a 6% eradication rate after sole debridement without implant exchange [7]. In line with the results of Jeys et al., many studies have shown the superior results of single-stage revision over sole debridement in patients with infected tumor prostheses [135]. Hardes et al. recommend one-stage revision surgery whenever possible and a two-stage revision if a one-stage revision is not feasible [46].

The results of debridement without implant removal for the treatment of infections in tumor prosthesis are highly varying. Moreover, there is no consensus on the number of debridements before staged revision surgery. Debridement may be considered in cases of acute infections caused by an antibiotic-sensitive pathogen; additional debridement can be repeated if infection is not completely resolved, but most likely further debridement will not be successful.

### 13.2. One-Stage Revision Surgery

There are only few high-quality studies reporting on the results of single-stage revision surgery for infected arthroplasties [7,46,47,83,90,108]. The concept of a single-stage revision arthroplasty following an infected total joint replacement was introduced by Buchholz et al. in the 1970s and entails a radical debridement and revision of all implants [136]. Based on the literature regarding periprosthetic infections after primary total hip and knee arthroplasty, the infection eradication rate is higher when all implants are removed in a one-stage or two-stage surgery. The degree of debridement and the subsequent decrease in the bioburden are associated with the successful resolution of these infections. In this setting, it is unclear whether all implants should be removed and revised during a one-stage revision surgery or whether some implants could be retained. Some surgeons perform an extensive soft tissue debridement and revise only the exchangeable and polyethylene implants during a single-stage revision surgery, without revising the anchoring components; however, most studied support that all implants should be removed during one-stage revision surgery [7,46,47,83,90,108]. If it is not possible to remove all implants, it is recommended that the retained implants should be thoroughly irrigated with various solutions, such as povidone-iodine or chlorhexidine. A pathogen should be isolated through joint aspiration prior to the one-stage revision surgery; otherwise, a two-stage revision surgery is recommended [137].

### 13.3. Two-Stage Revision Surgery

With a reported success rate of 63–100%, two-stage revision surgery is considered the most effective treatment strategy for infection in tumor protheses [7,8,10,90,132]. However, despite the large number of studies evaluating different treatment strategies for infections following tumor prostheses, the efficacy of DAIR vs. two-stage revision surgery has been compared in only few studies [8,10].

During a two-stage revision surgery, the implants are removed and replaced by antibiotic-loaded bone cement spacers for 2–3 months, and systemic antibiotics are administered during this period [46,47]. There is no agreed protocol regarding the choice and duration of antibiotics in two-stage revision protocols for infected tumor prostheses. Antibiotics are chosen based on culture results, while the duration is usually 3 months. In the second stage, the spacers are removed and replaced by the final implants [138]. Patients with enduring, high-grade infections, with pathogens that are resistant to antibiotics, or those who underwent unsuccessful one-stage revision surgery are recommended to undergo a two-stage revision surgery [46]. According to some studies, in the case of well-fixed cementless modular tumor prostheses, infected implants can be revised in two stages while keeping the anchorage stems [8,139]. Commercially available bone cements that contain antibiotics may be utilized for spacers, and a variety of additional antibiotics may be added given that they are heat-stable [140]. The drawbacks of a two-stage revision surgery include the long hospital stay, the increased bone loss and osteoporosis due to the lack of use, and the shortening of the affected limb [141,142].

The role of silver-coated prostheses in order to lower the infection risk in musculoskeletal tumor patients has not been extensively studied. Silver cations have antibacterial properties due to the disruption of DNA synthesis and of cellular membranes. A significant decrease in infection rate has been shown in several studies evaluating the use of silver-coated prostheses in tumor patients, compared to those receiving uncoated ones [143,144]. Wafa et al. also reported that the success rate of a two-stage revision surgery for infected silver-coated prostheses was higher compared to a two-stage revision surgery for an infected uncoated implant [144]. In line with the findings of Wafa et al., Zajonz et al. also reported that the reinfection rate following treatment of infected silver-coated protheses was lower compared to that of non-silver coated protheses (40% vs. 57%) [145].

## 14. Conclusions

The detrimental consequences of postoperative infections, especially in tumor patients, highlight the importance of prevention strategies in megaprosthetic reconstruction. A multidisciplinary approach between orthopedic surgeons and experts of different specialties is mandatory for the successful management of these infections [146]. The goals of treatment include infection resolution and limb salvation. Several different treatment strategies have been used for the management of megaprosthetic infections including irrigation and debridement with implant retention, one-stage revision surgery, and two-stage revision surgery; unfortunately, some patients end up with an amputated limb as a last resort. Two-stage revision has been shown to have the best functional outcomes and higher success rate and is currently considered the gold standard for the treatment of a chronic infection of a tumor prosthesis. However, in patients with isolated pathogens that are highly sensitive to antibiotics, one-stage revision surgery with or without exchange of the anchoring implants may be a viable alternative.

## Figures and Tables

**Figure 1 biology-12-00314-f001:**
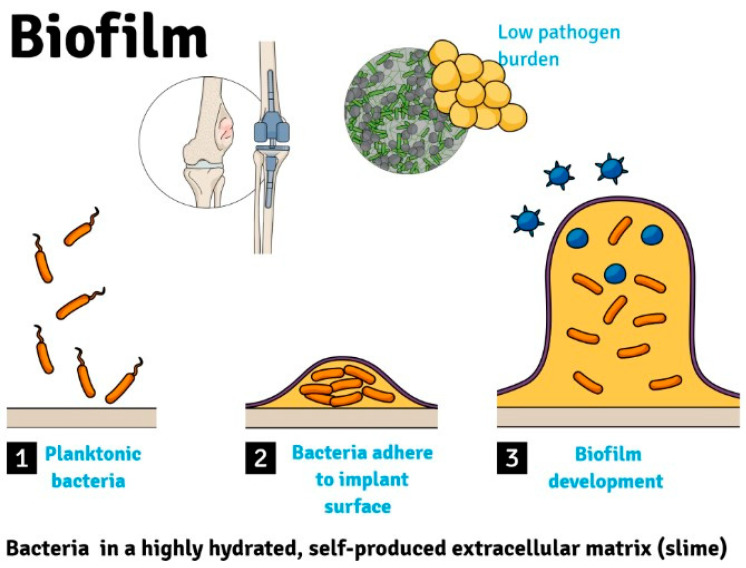
Bacteria adhere to implant surface to develop a biofilm in periprosthetic tumor infections.

**Table 1 biology-12-00314-t001:** Classification of infections in tumor prostheses.

Classification	Time Period
Class I (Early or acute)	Within 4 weeks following surgery
Class II	4 weeks to 2 years following surgery
Class III (Late or chronic)	>2 years

**Table 2 biology-12-00314-t002:** Diagnostic modalities for infections of tumor prostheses.

Modalities	Studies	Findings	Comments
Imaging	X-rays	Bone loss and implant loosening	Low sensitivity and specificity
	Bone scintigraphy Combined use of labeled leukocyte and Tc-99m sulfur colloid	High activity on labeled leukocyte. Low Activity on Tc 99m-sulfur colloid	Moderate sensitivity and specificity
	Ultrasound	Joint effusion or synovial hypertrophy	Operated-dependent, low specificity
	MRI	High signal intensity in T2 sequence	Usually not helpful due to artifacts
Laboratory	Blood workup	Elevated ESR, CRP	Commonly elevated during the first 2 weeks
		Elevated WBC count	Commonly elevated during the first 2 weeks
	Joint aspiration	Cultures	Sensitivity 70–90%, specificity 67–91%
		Histological examination	-
		Nucleic amplification techniques	PCR, NGS
	Implant Sonication	Cultures	-
		Nucleic amplification techniques	PCR, NGS
	Tissue biopsy	Cultures	Sensitivity 70–90%, specificity 67–91%
		Histological examination	-
		Nucleic amplification techniques	PCR, NGS

Abbreviations: MRI, Magnetic Resonance Imaging; ESR, erythrocyte sedimentation rate; CRP, C reaction protein; WBC, white blood cell; PCR, polymerase chain reaction; NGS, Next-Generation Sequencing.

**Table 3 biology-12-00314-t003:** Recommendations for perioperative use of antibiotics in conventional total joint arthroplasties.

Societies	Recommendations
International Consensus Meeting on Periprosthetic Infections [70]	One dose preoperatively and for the following 24 h postoperatively
Centers for Disease Control and Prevention [71]	One dose preoperatively without redosing
American Association of Hip and Knee Surgeons [69]	Postoperative antibiotics should be continued for 24 h
American Academy of Orthopaedic Surgeons [72]	Postoperative antibiotics should be continued for 24 h

## Data Availability

Data sharing is not applicable to this article.

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
