# Peer review of "Infections of Tumor Prostheses: An Updated Review on Risk Factors, Microbiology, Diagnosis, and Treatment Strategies"

_biology, 2023, doi:10.3390/biology12020314_

Round 1

Reviewer 1 Report

Reviewer report about manuscript by A.G. Tsantes and co-authors entitled ‘Infections of Tumor Prostheses: An updated review on micro-biology, risk factors, diagnosis, and treatment strategies’ submitted for publication to Biology journal

The evaluated manuscript is focused on a highly actual and complex medical problem of preventing bacterial infection associated with amputation of limbs and implantation of prostheses in patients with tumor lesions of bones. The team of authors of the article includes experienced surgical practitioners who have studied a significant amount of data on this problem, including data from meta-analysis of data collected in medical centers around the world. Despite the large amount of reliable data, the manuscript cannot be recommended for publication in the journal Biology for the following reasons:

1.      The review does not provide information on the dependence of the frequency of bacterial infection on the type of tumor, the nature of the operation performed and the type of prosthesis, although it is obvious that these factors significantly affect the risk of biofilms on the prostheses surface and complications caused by bacterial infection.

2.      The key point determining the approach to combating bacterial infection of prostheses is the origin of the causative agents (nosocomial, expansion of pathogens previously persisting in the patient or an accidental spill-over from the external environment). However, the authors entirely ignore molecular epidemiology of the pathogens found in the oncology patients with the prostheses. The problem of per-operation diagnosis of bacterial pathogens in the patients with tumors is not considered although it can substantially contribute to decreasing individual risk of the biofilm formation, purulent complications or poor engrafting of the prosthesis.

3.      Approaches to molecular typing of pathogens in connection with the problem under consideration have not been considered at all, although there are many problems in this area that affect the accuracy of diagnosis. The optimal methods of collecting biological samples for analysis and isolation of DNA from them for molecular typing should be considered.

4.      The description of the types of pathogens that cause the appearance of biofilms on the surface of prostheses and purulent complications in patients is too brief and does not fully reflect all possible options. For the S. aureus species, the affiliation of actual pathogens that caused massive complications to specific strains with known epidemiological characteristics should be considered, since knowledge of the molecular epidemiology of this species makes possible performing such an investigation. The term ‘coagulase-negative staphylococcus’ should be specified in more detail, since this group includes several species with significantly different epidemiological characteristics. There are both commensal species that are normal endosymbionts of human skin, and species that are detected in humans only at the time of acute infection. The term "group D Streptococcus" is outdated: the authors should use the name Enterococcus with an indication of a specific species. Enterobacteriaceae causing complications should also be characterized in more detail. Recommendations for using antibiotics of a certain type should be given in connection to the type of the bacterial pathogen. Types of drug resistance characteristic of them, for example, resistance to capbapenems in Klebsiella pneumoniae or methicillin in S. aureus should be taken into account.

5.      The list of references does not include articles newer than 2018, which is unacceptable for such a rapidly developing field as the diagnosis of bacterial infections using metagenomic technologies.

Taking into account the listed criticisms, I recommend the authors to carry out a deep revision of their text, and then re-send it for publication in the journal Biology.

Author Response

Reviewer 1

  1. The review does not provide information on the dependence of the frequency of bacterial infection on the type of tumor, the nature of the operation performed and the type of prosthesis, although it is obvious that these factors significantly affect the risk of biofilms on the prostheses surface and complications caused by bacterial infection.

Authors’ response: Although there is very limited information in the literature regarding the incidence of infections based on the type of the musculoskeletal tumor (i.e. whether osteoasarcorama is related to a higher infection risk compared to chondrosarcoma etc), a brief discussion in the “Risk factors” section of the manuscript is provided regarding this issue. Specifically, we state that the metastatic spread of the tumor and its histological characteristics were not found to be related to incidence of infections in most studies, while there is an ongoing debate about whether primary or metastatic lesions are more likely to develop infections. Regarding the nature of operation and the type of prosthesis, we discuss extensively in the manuscript that both of these parameters are associated with the infection risk. Specifically, we discuss in detail in theRisk factors” section of the manuscript that the location of the tumor and certain other characteristics such as the length of resection or the soft tissue resection affect the infection risk. The issue of type pf prosthesis, and more specifically whether cemented vs uncemented are associated with a higher infection risk is discussed in detail in the “Cemented vs uncemented fixation” subsection of the revised manuscript. Besides “cemented vs uncemented” fixation, other characteristics of implants have not been evaluated in the literature regarding their association with the infection risk.

  1. The key point determining the approach to combating bacterial infection of prostheses is the origin of the causative agents (nosocomial, expansion of pathogens previously persisting in the patient or an accidental spill-over from the external environment). However, the authors entirely ignore molecular epidemiology of the pathogens found in the oncology patients with the prostheses. The problem of per-operation diagnosis of bacterial pathogens in the patients with tumors is not considered although it can substantially contribute to decreasing individual risk of the biofilm formation, purulent complications or poor engrafting of the prosthesis.

Authors’ response: We agree with the reviewer that the origin of the periprosthetic infections is an important issue that needs to be discussed. Similar to all surgical site infections, the vast majority of periprosthetic infections are due to direct inoculation during surgery or through hematogenous spread from a distant infected focus. All prosthetic joints remain susceptible to hematogenous seeding from a distant primary focus during their entire indwelling time. High vascularity of periprosthetic tissue exposes the prosthesis to the highest risk of hematogenous infection in the first years after implantation. Moreover, the primary infected focus in case of hematogenous spread depends on the specific pathogen (i.e. Viridans streptococcci cause dental infections etc.). This issue is now discussed in the “Introduction” section of the manuscript, while also more details are given regarding the origin of specific pathogens in the “Pathophysiology” section of the revised manuscript.

  1. Approaches to molecular typing of pathogens in connection with the problem under consideration have not been considered at all, although there are many problems in this area that affect the accuracy of diagnosis. The optimal methods of collecting biological samples for analysis and isolation of DNA from them for molecular typing should be considered.

Authors’ response: We agree with the reviewer that the molecular approach for the laboratory diagnosis of periprosthetic joint infections is of great significance and should be discussed in the manuscript. A paragraph with a detailed description of the molecular biology techniques to detect tumor implant-related infections has been added in the “Diagnosis” section of the revised manuscript.

  1. The description of the types of pathogens that cause the appearance of biofilms on the surface of prostheses and purulent complications in patients is too brief and does not fully reflect all possible options. For the aureus species, the affiliation of actual pathogens that caused massive complications to specific strains with known epidemiological characteristics should be considered, since knowledge of the molecular epidemiology of this species makes possible performing such an investigation. The term ‘coagulase-negative staphylococcus’ should be specified in more detail, since this group includes several species with significantly different epidemiological characteristics. There are both commensal species that are normal endosymbionts of human skin, and species that are detected in humans only at the time of acute infection. The term "group D Streptococcus" is outdated: the authors should use the name Enterococcus with an indication of a specific species. Enterobacteriaceae causing complications should also be characterized in more detail. Recommendations for using antibiotics of a certain type should be given in connection to the type of the bacterial pathogen. Types of drug resistance characteristic of them, for example, resistance to capbapenems in Klebsiella pneumoniae or methicillin in S. aureus should be taken into account.

Authors’ response: We agree with the reviewer that a more detailed description of the microbiology of these infections is valuable to the readers. Based on the reviewer’s recommendation, a new section entitled “Pathophysiology” has been added to the manuscript including a more detailed description of the causative pathogens for tumor implant-related infections. Moreover, the “group D Streptococcus” has been replaced by “Enterococcus spp” across the manuscript.

  1. The list of references does not include articles newer than 2018, which is unacceptable for such a rapidly developing field as the diagnosis of bacterial infections using metagenomic technologies.

Authors’ response: We agree with the reviewer, therefore the manuscript and the reference list have been revised, including a detailed discussion regarding molecular biology techniques along with relevant recent references.

Reviewer 2 Report

This review article describes the current knowledge regarding the epidemiology, diagnosis and different risk factors for developing infections of tumor prostheses. Various factors that can affect the occurrence of infection are described, including Duration of surgery, Antibiotic prophylaxis, Chemotherapy and Radiation therapy, Neutrophil count, use of Surgical drains and type of fixation. Different surgical options for resolving complications are presented.

I consider this review article of high quality with an extensive list of literature reviewed and significant contribution to the field. I suggest some minor corrections:

INTRODUCTION:

Please add the reference for the paragraph beginning with “Infections of tumor prostheses are often associated with…”.

CLASSIFICATION

Please add the reference for the paragraph beginning with “Early postoperative and hematogenous infections are generally…”.

DIAGNOSIS

Table 2: If reporting different studies in Table 2, references should be included.

Please add the reference for the paragraph beginning with “A combination of clinical, histological, and microbiological parameters…”.

ANTIBIOTIC PROPHYLAXIS

Table 3: add the references of the Societies and Recommendations in Table 3.

Please add the reference for the paragraph beginning with “Intraoperative redosing is usually recommended…”.

Author Response

Reviewer 2

I consider this review article of high quality with an extensive list of literature reviewed and significant contribution to the field.

Authors’ response: We thank the reviewer for his/her kind words.

INTRODUCTION:

Please add the reference for the paragraph beginning with “Infections of tumor prostheses are often associated with…”

Authors’ response: Based on the reviewer’s recommendation, references have been added in the respective paragraph.

CLASSIFICATION

Please add the reference for the paragraph beginning with “Early postoperative and hematogenous infections are generally…”.

Authors’ response: Based on the reviewer’s recommendation, references have been added in the respective paragraph.

DIAGNOSIS

Table 2: If reporting different studies in Table 2, references should be included.

Authors’ response: Table 2 does not report different studies.

Please add the reference for the paragraph beginning with “A combination of clinical, histological, and microbiological parameters…”.

Authors’ response: Based on the reviewer’s recommendation, references have been added in the respective paragraph.

ANTIBIOTIC PROPHYLAXIS

Table 3: add the references of the Societies and Recommendations in Table 3.

Authors’ response: Based on the reviewer’s recommendation, the respective references have been added in Table 3.

Please add the reference for the paragraph beginning with “Intraoperative redosing is usually recommended…”

Authors’ response: Based on the reviewer’s recommendation, references have been added in the respective paragraph.

Reviewer 3 Report

The authors have thoroughly documented the microbiology, risk factors, diagnosis and treatment associated with infections of tumor prostheses. The pathogenesis of infections in tumor prosthesis is linked to several bacterial infections (Staphylococcus spp.,). Early detection is increasingly difficult due to varying clinical symptoms and due to lack of diagnostic tools. The authors highlighted these concepts very well documented. The structure of the review article talks about recent developments and concepts to understand bacterial infection, epidemiology, diagnosis, and risk factors associated with tumor prostheses. Recommendations for the use of antibiotics and problems associated with chemotherapy and radiation therapy are well explained.

The manuscript is very well written, and the conclusions were correlated with the literature review findings. Overall, I feel this is a very good literature review of the study and I have no further comments.

Author Response

Reviewer 3

The authors have thoroughly documented the microbiology, risk factors, diagnosis and treatment associated with infections of tumor prostheses. The pathogenesis of infections in tumor prosthesis is linked to several bacterial infections (Staphylococcus spp.,). Early detection is increasingly difficult due to varying clinical symptoms and due to lack of diagnostic tools. The authors highlighted these concepts very well documented. The structure of the review article talks about recent developments and concepts to understand bacterial infection, epidemiology, diagnosis, and risk factors associated with tumor prostheses. Recommendations for the use of antibiotics and problems associated with chemotherapy and radiation therapy are well explained.

The manuscript is very well written, and the conclusions were correlated with the literature review findings. Overall, I feel this is a very good literature review of the study and I have no further comments.

Authors’ response: We thank the reviewer for his/her kind words.

Reviewer 4 Report

The authors in this review elucidated the current knowledge regarding the epidemiology, risk factors, microbiology, and diagnosis of infections of tumor prostheses, which well written review. However, I suggest below:

The author elucidates the role of antibiotic prophylaxis (long term and short term), it is also good to add the effectiveness of single dose and double dose of prophylactic antibiotics.

(Effectiveness of Single vs Multiple Doses of Prophylactic Intravenous Antibiotics in Implant-Based Breast Reconstruction) A Randomized Clinical Trial. Jessica Gahm et al.

Author Response

Reviewer 4

The author elucidates the role of antibiotic prophylaxis (long term and short term), it is also good to add the effectiveness of single dose and double dose of prophylactic antibiotics. (Effectiveness of Single vs Multiple Doses of Prophylactic Intravenous Antibiotics in Implant-Based Breast Reconstruction) A Randomized Clinical Trial. Jessica Gahm et al.

Authors’ response: Based on the reviewer’s recommendation a short discussion about the effectiveness of single dose vs double dose of prophylactic antibiotics has been added in the “Antibiotic prophylaxis” section of the  revised manuscript, including the proposed reference.